# Two Years’ Experience of Intensive Home Hemodialysis with the Physidia S^3^ System: Results from the RECAP Study

**DOI:** 10.3390/jcm12041357

**Published:** 2023-02-08

**Authors:** Hafedh Fessi, Philippe Nicoud, Tomas Serrato, Olivia Gilbert, Cécile Courivaud, Salima Daoud, Marion Morena, Michel Thomas, Bernard Canaud, Jean-Paul Cristol

**Affiliations:** 1Nephrology Department, Tenon Hospital, 75020 Paris, France; 2Dialysis Department, Aural Dialysis Center, 69008 Lyon, France; 3Emergency Department, Léman Hospital, 74200 Thonon Les Bains, France; 4Manhès Hospital Center, 91700 Fleury-Mérogis, France; 5Charles Mion Foundation, AIDER-Santé, 34090 Montpellier, France; 6Nephrology Department, University Hospital Center of Besancon, 25000 Besancon, France; 7Monitoring Force Group, 78600 Maisons-Laffitte, France; 8PhyMedExp, University of Montpellier, INSERM, CNRS, Department of Biochemistry and Hormonology, University Hospital Center of Montpellier, 34000 Montpellier, France; 9Physidia, 49124 Saint-Barthélemy-d’Anjou, France; 10Faculty of Medicine, University of Montpellier, 34000 Montpellier, France

**Keywords:** intensified hemodialysis, slow daily dialysis, dialysis efficiency, adequacy, dialysis quantification

## Abstract

The RECAP study reports results and outcomes (clinical performances, patient acceptance, cardiac outcomes, and technical survival) achieved with the S^3^ system used as an intensive home hemodialysis (HHD) platform over a three-year French multicenter study. Ninety-four dialysis patients issued from ten dialysis centers and treated more than 6 months (mean follow-up: 24 months) with S^3^ were included. A two-hour treatment time was maintained in 2/3 of patients to deliver 25 L of dialysis fluid, while 1/3 required up to 3 h to achieve 30 L. The additional convection volume produced by means of the SeCoHD tool (internal filtration backfiltration) was 3 L/session, and the net ultrafiltration produced to achieve dry weight was 1.4 L/session. On a weekly basis, an average 156 L of dialysate corresponding to 94 L of urea clearance when considering 85% dialysate saturation under low flow conditions was delivered. Such urea clearance was equivalent to 9.2 [8.0–13.0] mL/min weekly urea clearance and a standardized Kt/V of 2.5 [1.1–4.5]. The predialysis concentration of selected uremic markers remained remarkably stable over time. Fluid volume status and blood pressure were adequately controlled by means of a relatively low ultrafiltration rate (7.9 mL/h/kg). Technical survival on S^3^ was 72% and 58% at 1 and 2 years, respectively. The S^3^ system was easily handled and kept by patients at home, as indicated by technical survival. Patient perception was improved, while treatment burden was reduced. Cardiac features (assessed in a subset of patients) tended to improve over time. Intensive hemodialysis relying on the S^3^ system offers a very appealing option for home treatment with quite satisfactory results, as shown in the RECAP study throughout a two-year follow-up time, and offers the best bridging solution to kidney transplantation.

## 1. Introduction

Intensive home hemodialysis (IHHD) [1] is recognized as delivering a more physiological and efficient renal replacement therapy in chronic kidney disease patients with better outcomes [2,3,4,5]. IHHD remains underutilized as compared to in-center hemodialysis for various reasons, which include technical complexity, fears of home and selfcare treatment, and poor perception about this modality. Over the last decade, technical advances in manufacturing dialysis machines have considerably reduced technical complexity, facilitated home implementation, and improved patient perception toward this modality.

The NxStage system has paved the way and opened a new therapeutic avenue in the US and then abroad by confirming its safety and attractiveness and then by showing clinical benefits [6,7,8]. Such progresses have been made possible by the development of dedicated home HD machines featuring more friendly user interfaces and web-based connection, facilitating handling, suppressing needs for water treatment systems via simplified dialysate delivery systems, and finally providing a plug and play approach for HD technology [9,10,11,12].

The Physidia S^3^ system was developed in France few years later with the aim of further reducing the burden of home HD treatment both for patients and care providers by keeping the concept of more intensive dialysis relying on a low flow and more frequent approach [13].

In this study, we aimed to report results and outcomes (clinical performances, patient acceptance, cardiac outcomes, and technical survival) achieved with the S^3^ system used as an intensive home HD platform over a three-year French multicentric study.

## 2. Materials and Methods

### 2.1. Description of the Physidia S^3^ System

The Physidia S^3^ system (named the S^3^ system) is a bagged delivery cycler and portable communicating dialysis monitoring device specifically designed for short daily hemodialysis (HHD) and self-administered treatment [13]. The S^3^ device has a compact and portable cubic design (dimension 40 × 40 × 40 cm), weighing less than 25 kg and being specifically designed for short daily low-dialysate-flow HHD (dialysate flow rates of 150 to 200 mL/min).

The dialysis fluid is stored in 5 L sterile bags. Dialysis fluid bags are stored in a plastic shelf with 5 to 7 racks. The balance chamber technology with an adjustable ultrafiltration rate is able to provide substantial additional convective volume to weight loss by enhancing internal filtration. The S^3^ system is not bound to a specific hemodialyzer choice. The S^3^ system has a cartridge setup, automated prime function, removable tablet with touch-screen patient interface, and the ability to transmit treatment data to a physician at the end of the dialysis session via a tablet connected by Bluetooth to the monitor. The S^3^ system was approved and introduced on the European market (CE mark) in 2013.

### 2.2. Study Design

RECAP is a retrospective longitudinal multicenter study conducted in France. Ten French dialysis facilities, which are experts in home dialysis treatment, collaborated in this observational study to assess outcomes associated with IHHD using the Physidia S^3^ system (Physidia SAS, Saint-Barthélémy-d’Anjou, France). IHHD refers in this case to a short (≥2 h) and more frequent weekly treatment schedule (≥5 sessions/week). The treatment schedule prescription (treatment time, frequency, and dialysate volume per session and week) was defined by the referent nephrologist in each dialysis facility according to patient needs and tolerance. Patient monitoring and treatment efficacy were led by the referent nephrologist based usually on a quarterly outpatient clinic visit capturing patient key parameters including vital signs, laboratory tests, and operating dialysis conditions. Data captured during this visit were sent to the patient’s electronic medical record, anonymized, and shared in a central data repository system.

The RECAP study complied with the General Data Protection Regulation (RGPD) and was conducted in agreement with French legislation on non-interventional studies (reference INDS: MR 3309311018).

### 2.3. Patients

Ten French dialysis and training dialysis centers were selected, in which eligible patients older than 18 years were screened. Ninety-four patients having started IHHD between 1 January 2015 and 30 June 2018 and treated more than 6 months and up to 42 months’ follow-up (mean follow-up was 24 months) with the S^3^ system were included in the analysis. All patients were provided an information sheet at least one month prior to the start of the study. Patients who expressed their opposition to data collection were not included in the study. A flow chart of the study population is presented in Figure 1.

### 2.4. Data Collection

Data were collected retrospectively on a quarterly basis from the first day of use at home with the S^3^ system until 30 September 2018, or until the last day of use of the S^3^ system in the case of permanent discontinuation before the end of the study follow-up. Socio-demographic data, comorbidities, disabilities, status before start of IHHD, previous renal replacement therapy modalities, biological data, and dialysis prescriptions were captured from the electronic Case Report Form (eCRF). Blood pressure before, during, and after the dialysis session were extracted directly from the dialysis monitor session reports; anonymized data were recorded and centralized on a regular basis by Physidia company. The identification number of the monitor S^3^ and the date of session were used to match the eCRF data with the S^3^ monitor data.

### 2.5. Calculation and Statistical Analysis

Continuous data were expressed as mean values ± standard deviation (SD), and medians with 25th and 75th percentiles or IQR were calculated for non-normally distributed data. Only dialysis session reports lasting 90 min or more were included in the analysis.

Vital parameters, including body weight, systolic and diastolic blood pressure (SBP and DBP), dialysis parameters, and biochemical tests were captured at baseline and for every 3-month period until 24 months of follow-up (due to the small number of patients with a follow-up of more than 24 months).

The dialysis dose delivered used as a surrogate marker of clinical performances was calculated. The urea reduction rate (URR) was used to assess the efficacy of the dialysis session over time. In addition, the standard weekly urea Kt/V and the weekly urea clearance equivalent were estimated from blood and dialysate flow using the equation of Michaels adapted by Leypoldt [14,15]. According to this equation, urea dialysate saturation (D/P) reaches 90% for dialysate and blood flows of 180 and 250 mL/min, respectively. Based on this assumption, one may estimate the weekly urea clearance as 0.9 of the total dialysate volume delivered per week. The weekly standard Kt/V was estimated as the ratio of weekly urea clearance and total body water. Total body water was estimated at 0.55 of dry weight for male and 0.50 for female patients.

For each parameter, a mixed linear model for repeated measures with a random intercept for each patient was used to assess the statistical significance of the evolution over time. The model was adjusted for time and gender as fixed effects and center as the random effect.

Some biomarkers benefited from additional transformation. For iron markers, namely ferritin and transferrin coefficient saturation, the Napierian logarithm value was used in the model. For bone mineral disorder biomarkers, tertile classes were used to assess the evolution of parathyroid hormone (PTH) (low (<130 pg/mL), normal (130–585 pg/mL), and high (>585 pg/mL)) and phosphorus (low (<0.9 mmol/L), normal (0.9–1.4 mmol/L), and high (>1.4 mmol/L)).

An intradialytic hypotension episode was defined as any SBP drop < 90 mmHg measured during the HD session [16].

Technical survival on the S^3^ system at 1 and 2 years was estimated using the Kaplan–Meier method. The failure of the method was estimated from the cumulative incidence of returning in-center using a competitive risk method with kidney transplantation as the main competitive risk.

No replacement of the missing values was carried out.

All statistical analyses were performed with SAS software^®^ version 9.4 and SAS/STAT 15.1 on Windows (SAS Institute, Cary, NC, USA).

## 3. Results

### 3.1. Patient’s Characteristics

Ninety-four patients who fulfilled selection criteria were included in the final analysis.

Patients’ characteristics upon inclusion are displayed in Table 1. The mean age was 49.7 ± 14.9 years old, and 64% of patients were male. The mean predialysis body weight was 75 kg and 64 kg in males and females, respectively. The mean body mass index was 25.3 ± 5.1 kg/m². While most patients had a normal body mass (54.2%), 31.3% of patients were overweight and 13.3% obese. The mean Charlson comorbidity score was 3.6 ± 2.0 [range 1–12]. Time spent on dialysis treatment before entering S^3^ treatment was 12.2 months [3.2–47.0]. As shown, 79.8% of patients were treated in-center, and 11% were incident-naive dialysis patients. Overall, 40.4% of patients had at least one kidney transplant, and the latest kidney transplantation was 12 ± 7.4 years ago. Through the follow-up period, 27,826 dialysis sessions were extracted from the database and analyzed.

### 3.2. Renal Replacement Treatment Prescription and Operating Conditions

The main features of dialysis prescription and operating conditions are displayed in Table 2. The dialysis treatment time was 133 min [120–180] and 129 min [120–150] for female and male patients, respectively. The vascular access used was an arteriovenous fistula in 97.9% of patients. One patient used two vascular access types during the entire study. Access cannulation relied on the buttonhole method in 89.2% of patients. Two patients were treated by means of a tunneled central venous catheter. Training time to home installation was 6.9±3.9 weeks including punction training.

All patients received five sessions or more per week, while two-thirds of them received six sessions weekly, then being considered as a daily treatment (Table 3). A two-hour treatment time was delivered in two-thirds of patients, while one-third received up to 3 h. The dialysis fluid volume exchanged was 25 L per session in three-quarters of patients, while one-quarter received up to 30 L per session. The additional convection volume produced by means of the SeCoHD tool (internal filtration back filtration) was 3 L per session, and the net ultrafiltration produced to achieve dry weight was 1.4 L per session, which translated to a normalized ultrafiltration rate of 7.9 mL/h/kg [range 2–10]. In brief, the RECAP study delivered, on a weekly basis, an average of 156 L of dialysis fluid corresponding to 94.2 L of urea clearance when considering that the dialysate saturation achieved was 85% in this low flow condition. This is equivalent to a weekly urea clearance of 9.17 ± 0.81 mL/min and a standardized weekly Kt/V of 2.52 ± 0.67.

### 3.3. Clinical Performances and Laboratory Data

Fluid volume management and blood pressure control are presented in Figure 2 and Figure 3. The dry weight decreased initially (first 6 months) by 0.5 kg, then slowly went back to initial values and remained relatively constant over the observation time (median value 71.5 kg). The pre-dialysis systolic blood pressure (SBP) was significantly higher in men than in women (*p* = 0.031) at baseline, remaining relatively stable over time with the same difference (*p* = 0.9134). Pre-dialysis diastolic blood pressure (DBP) did not differ according to gender at baseline (*p* = 0.3499), but interestingly it decreased during the first 6 months (*p* = 0.0071) and then stabilized until the end of follow-up (Figure 2 and Figure 3).

The urea reduction rate (URR) ranged from 45.8% to 57.4% throughout the follow-up with a P25 that ranged from 5.9% to 50% and P75 that ranged from 52% to 68.2%. The estimated weekly standard urea Kt/V was 2.40 [2.0–3.2]. The delivered urea clearance equivalent per week was 9.0 mL/min [8,9,10] (Table 3).

Plasma phosphate (Figure 4) and PTH concentrations remained relatively stable during the 24-month follow-up period, indicating that divalent ions and bone mineral disorders were adequately controlled (Table 4). The total serum calcium increased from baseline to the 9–12-month period and then decreased until the end of the 24 months of follow-up, reaching statistical significance (*p* = 0.0497).

The plasma creatinine levels did not change over time, suggesting that creatinine clearance and muscle mass remained constant. Plasma β2-microglobulin levels were documented in a subset of patients (27 patients at baseline) and remained stable over time with a value ≤ 25 mg/L (Table 4).

Anemia and iron status control were satisfactory. As shown, the behavior of hemoglobin levels followed a two-phase change: hemoglobin decreased in the initial phase (11.38 ± 1.61 g/dL at baseline to 10.53 ± 1.52 g/dL at 3–6 months period), then hemoglobin increased back to reach 11.33 ± 1.54 g/dL at the end of the 24-month follow-up (Table 4). Ferritin levels followed the same behavior; they decreased initially (9–12 months) and then improved subsequently, reflecting IV iron supplementation. Finally, serum albumin concentrations did not change significantly, while they tended to increase over time (*p* = 0.037) (Figure 5).

### 3.4. Cardiac Morphologic Features

In a subset of 20 patients, cardiologic features were more precisely monitored via echocardiography during three months of follow-up. As shown in Table 5, the left ventricular mass reduced by 9.6% at 3 months (103 vs. 114 g/m²), while the ejection fraction remained in the normal range and did not change over time (62.0% vs. 61.5%).

### 3.5. Technique Survival

The probability of permanently continuing HHD with the S^3^ monitor at 1 and 2 years was 72% and 58%, respectively. The probability of the continued use of HHD with the S^3^ monitor during follow-up, estimated using the same method, is displayed in Figure 6.

The main reason for the permanent discontinuation of HHD with the S^3^ monitor was kidney transplantation (n = 19) followed by transfer to in-center hemodialysis (n = 11).

The main causes of transfer to in-center HD were the loss of a caregiver (mandatory in France) and the patient’s loss of autonomy, requiring treatment. Other more minor causes of discontinuation of treatment with the S^3^ system were burnout, lack of patient compliance, or referral to another home dialysis technique.

### 3.6. Evolution of the Professional Situation

The evolution of the professional situation is displayed in Figure 7.

Among the 11 unemployed patients at baseline, a professional activity had been resumed during the follow-up period by 3 patients. In total, 16 patients (8 men and 8 women) had a period of unemployment during follow-up, and 25% of these patients returned to work at least once during the study period.

## 4. Discussion

The RECAP is a two-year retrospective cohort study reporting the results and patient experience of 94 kidney disease patients who received IHHD (i.e., short daily, low flow) with the Physidia S^3^ system. The RECAP study confirms the clinical benefits of IHHD and enlarges our knowledge by analyzing a relatively large dialysis population presenting with various clinical and metabolic profiles having used the same S^3^ device with a similar protocol, reducing care variations and then providing homogeneous findings [3,4,5,17,18,19].

Intensive home hemodialysis stands, in this study, for more frequent (≥5 sessions weekly), short sessions (120 min) and a low dialysis fluid volume (25 L) [19]. As indicated, the treatment schedule was well-accepted and able to deliver an adequate renal replacement treatment at home in the vast majority of patients. Technical survival on the same treatment schedule using the S^3^ system was 72% and 58%, at 1 and 2 years, respectively. Interestingly, the main reason for discontinuing home treatment with the S^3^ system was kidney transplantation in 21% of patients [5]. In this case, stopping home HD should be considered as a success in the management of a renal patient’s trajectory, suggesting that a hemodialysis cycler specifically designed for home treatment on a plug and play concept offers the best bridging solution to kidney transplant [5]. Other reasons for stopping home treatment with the S^3^ system and returning to in-center hemodialysis represent home treatment failure but not a technical failure.

A renal replacement treatment schedule was maintained and relatively unchanged over time. All patients received five sessions or more per week, and among them two-thirds received six sessions weekly, then being considered as a daily treatment. A two-hour treatment time was maintained in two-thirds of patients to deliver 25 L of dialysis fluid, while one-third required more up to 3 h to achieve a 30 L dialysis fluid exchange. The additional convection volume produced by means of the SeCoHD tool (internal filtration back filtration) was 3 L per session in addition to the net ultrafiltration produced to achieve dry weight, which was 1.4 L per session. In brief, the RECAP study delivered, on a weekly basis, an average 156 L of dialysate corresponding to 94 L of urea clearance when considering 85% dialysate saturation at low flow conditions. Such urea clearance is equivalent to 9.2 [range 8.0–13.0] mL/min weekly urea clearance and a standardized Kt/V of 2.52 [range 1.1–4.5] [15,20]. An estimate of dialysis dose delivered in the RECAP study addressed most of the patients’ metabolic needs and compares favorably with recent studies reporting on short daily and low dialysis fluid volume. This interesting finding may be partially explained by the additional convective dose (+3 L per session corresponding to 15 to 18 L per week) provided through the SeCoHD option.

Fluid volume management and blood pressure control were achieved adequately. Dry weight was maintained over time with rare or timely appropriate weight loss adjustment. Blood pressure was maintained in a narrow target range, as shown by the predialysis mean systolic and diastolic blood pressure values of 140 [135–145] mmHg and 80 [75–85] mmHg, respectively. No significant differences were noted between male and female patients. Hemodynamic stability and dialysis tolerance were quite satisfactory, as highlighted by the very low incidence of intradialytic hypotension and almost no serious intradialytic morbidity. These findings are likely due to the low ultrafiltration rate of 7.9 mL/h/kg [range 2–10], resulting in better hemodynamic stability, while a sustained volume and sodium depletion might be achieved over time [21].

Solute removal capacity and uremic control were satisfactorily achieved with intensive hemodialysis low dialysate volume, as highlighted by various indicators of dialysis efficacy used in this study. The urea reduction rate per session was over 50%, corresponding to a weekly standard Kt/V of 2.5 [20,22]. The predialysis concentration of selected uremic markers remained remarkably stable over time: urea concentration was lower than 25 mmol/L; β2-microglobulin concentration was lower than 25 mg/L; phosphate concentration was lower than 1.55 mmol/L. Interestingly, circulating levels of selected uremic markers remained in target while dietary protein intake was preserved. Bone mineral disorder markers (plasma calcium and PTH) were also controlled adequately to maintain in target. Anemia correction, as shown by hemoglobin concentrations, was maintained between 11 and 12 g/dL. Nutritional status was well preserved, as indicated by key parameters such as albumin, which was maintained between 40 and 41 g/dL. Electrolytic disorders were well controlled in the study, as indicated by relatively stable concentrations of plasma bicarbonate. The individualized prescription of dialysate potassium and bicarbonate may have facilitated this endeavor.

Patients’ perspectives also appeared to be improved throughout the study follow-up period. The patient perception and satisfaction survey previously reported by our group confirmed that the S^3^ system was easy to use, easily implemented at home, and facilitated patients’ mobility [13]. Despite the short daily treatment schedule, the S^3^ system was considered as a less intrusive treatment modality than previously experienced with in-center hemodialysis treatments. In addition, a slow flow treatment system tends to reduce the dialysis burden by preventing intradialytic morbidity and shortening postdialysis recovery time. In line with these positive clinical effects, almost 25% of unemployed patients tended to return to work during the two-year follow-up period.

In agreement with recent daily dialysis studies, cardiac features as assessed by echocardiography were improved with intensive dialysis at home [23,24,25,26]. As shown in a subgroup of patients, left ventricular mass was reduced by 9.6% at 3 months, while the ejection fraction remained remarkably stable in the normal range of 61 to 62%. This finding indicates that cardiac remodeling tends to improve with intensive dialysis, while cardiac function is better preserved. This observation is perfectly in line with the frequent hemodialysis study.

Several advantages of intensive home dialysis therapy relying on the S^3^ system with a bagged cycler system are illustrated in this study: on one hand, the low flow concept permits one to saturate dialysate (up to 75 and 80%) with uremic compounds and then to reduce dialysis volume consumption and save water and power; on the other hand, the friendly interface, fine edge design, and compacity of the S^3^ system facilitate handling operations and reduce the intrusiveness of home treatment.

The study also has some limitations that deserve to be explored in further clinical studies. Firstly, RECAP is a retrospective clinical study and not a prospective one. Secondly, the number of patients included is relatively low, although not negligible if we take into account the low proportion of patients treated using frequent hemodialysis at home in France. Thirdly, this study is limited to France. Fourth, the outcome (clinical/technique) failure should have been compared with other HHD programs that use conventional HD machines. Unfortunately, such a comparison was not possible in the reported study, since patients treated with Physidia S^3^ did not experience a home HD thrice-weekly treatment schedule with conventional in-center HD material. From this perspective, one can only compare our long-term technical survival rate (52%) to previous studies [27,28] reporting higher technical acceptance (70 to 88%) based on a thrice-weekly treatment schedule and using conventional in-center HD material. Such a discrepancy likely reflects the burden of daily treatment and not home treatment with the Physidia S^3^ device.

## 5. Conclusions

Intensive hemodialysis relying on the S^3^ system offers a very appealing option for home treatment with quite satisfactory results, as shown in the RECAP study throughout a two-year follow-up time. The S^3^ system was easily handled and adopted by patients at home. Overall, the results confirmed that a more frequent, short and low flow treatment schedule may deliver adequate treatment that fits with the majority of dialysis patients’ needs. The most prominent findings are as follows: firstly, patient perception was improved, while treatment burden was reduced despite a more frequent treatment schedule; secondly, cardiac features tended to improve over time, confirming that intensive hemodialysis has the capacity to reduce cardiac health in dialysis patients; thirdly, home treatment based on the S^3^ system offers the best bridging solution to kidney transplantation.

## Figures and Tables

**Figure 1 jcm-12-01357-f001:**
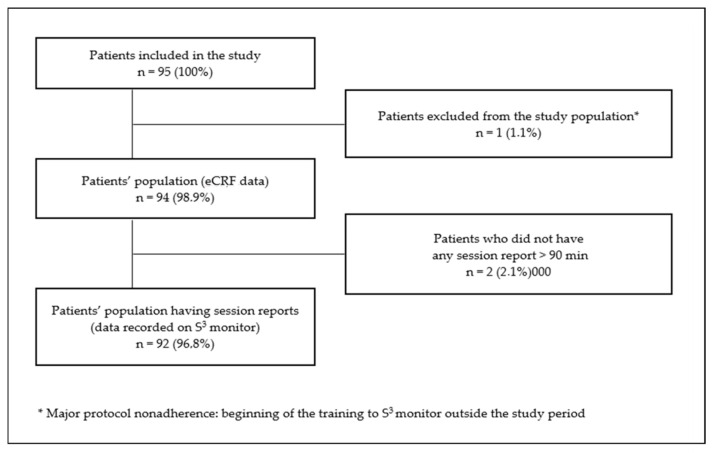
Flowchart of the study population.

**Figure 2 jcm-12-01357-f002:**
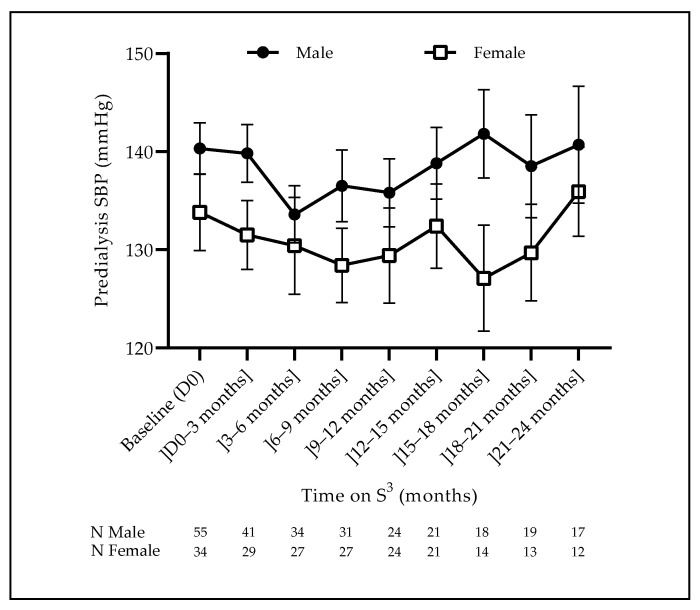
Evolution of the predialysis systolic blood pressure (SBP) (mean ± SEM).

**Figure 3 jcm-12-01357-f003:**
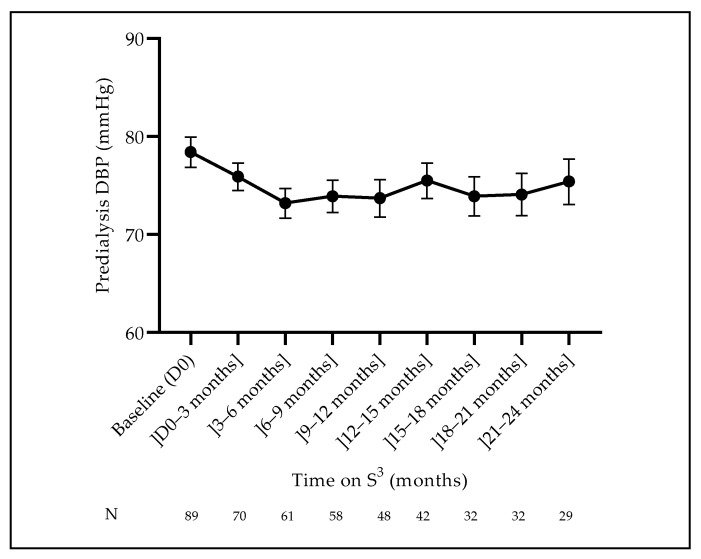
Evolution of the predialysis diastolic blood pressure (DBP) (mean ± SEM).

**Figure 4 jcm-12-01357-f004:**
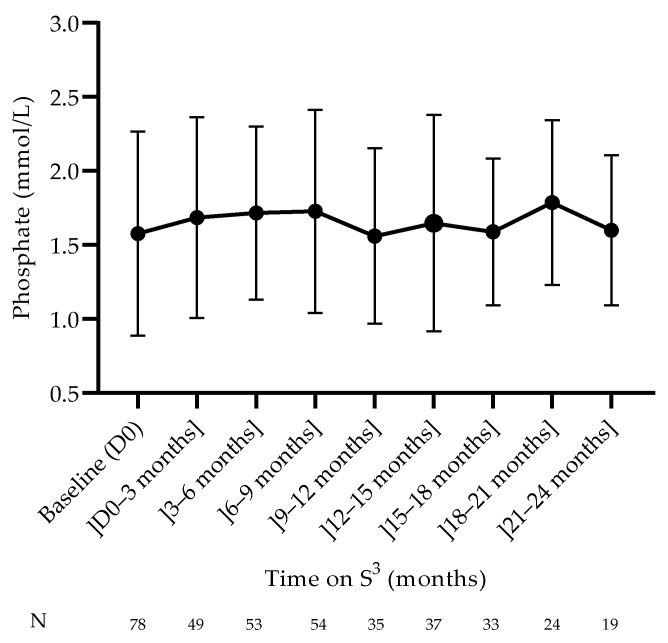
Evolution of plasma phosphate concentrations (median ± IQR).

**Figure 5 jcm-12-01357-f005:**
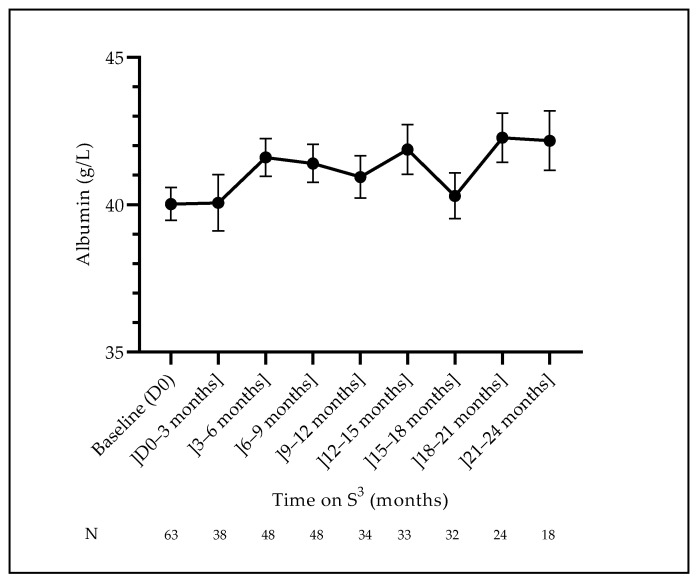
Evolution of plasma albumin concentrations (mean ± SEM).

**Figure 6 jcm-12-01357-f006:**
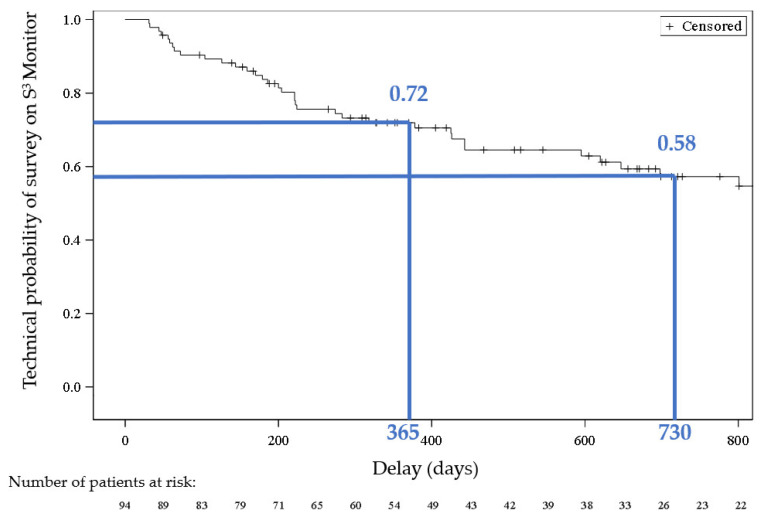
Technical survival probability of continuing HHD with S^3^ monitor during the follow-up period.

**Figure 7 jcm-12-01357-f007:**
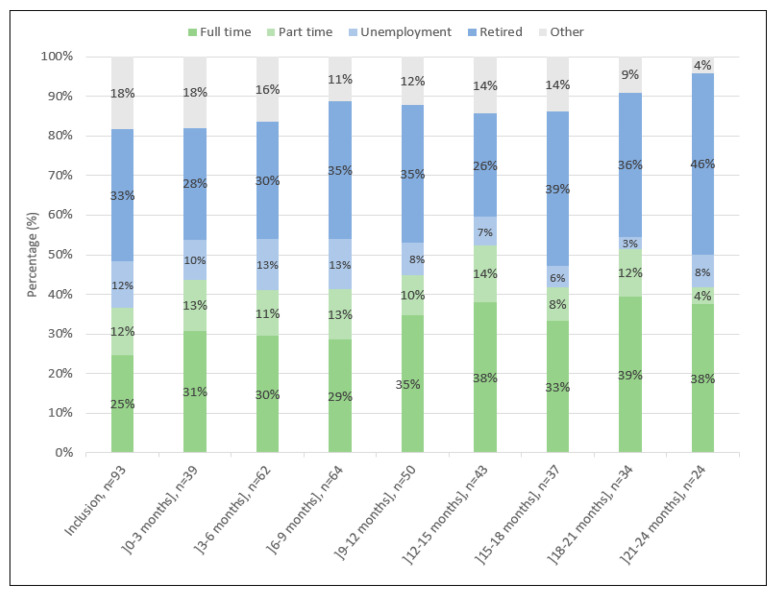
Evolution of the professional situation.

**Table 1 jcm-12-01357-t001:** Patient characteristics.

Category	Values
Patient Characteristics (n = 94)	
Gender, male	60 (63.8)
Age (years)	49.7 ± 14.9
Predialysis Weight (kg)	71.5 (20): M 75 (23.3); F64 (16.1)
Weight Loss (mL/ses.)	1393 (654): M 1458 (736); F1293 (539)
Body Mass Index (kg/m^2^)	25.3 ± 5.1
Underweight	1 (1.2)
Normal	51 (54.2)
Overweight	29 (31.3)
Obese	13 (13.3)
Charlson comorbidity score (points)	3 (2–4)
Professional situation at inclusion (n = 93)	
Full-time	23 (24.7)
Part-time	11 (11.8)
Unemployment	11 (11.8)
Retired	31 (33.3)
Other	17 (18.3)
Primary Kidney Disease * (n = 94)	
Chronic Glomerulonephritis	19 (22.4)
Chronic Interstitial Nephritis	5 (5.9)
Autosomal Polycystic Kidney Disease	15 (17.6)
Diabetic Kidney Disease	4 (4.7)
Hypertensive Kidney Disease	11 (12.9)
Vascular Kidney Disease	4 (4.7)
Other	36 (42.4)
Prior renal replacement modality (n = 94)	
In-center hemodialysis (3 × 4 h/week)	75 (79.8)
Peritoneal dialysis	2 (2.1)
Kidney transplant	3 (3.2)
Incident ESKD	10 (10.6)
HHD with another machine than S^3^ monitor	4 (4.3)
Prior dialysis duration (months) ** (n = 84)	12.2 (3.2–47)
Number of previous transplants (n = 94)	
0	56 (59.6)
1	27 (28.7)
2	11 (11.7)

Values are expressed as n (%), mean ± SD, median (IQR). Abbreviation: ESRD, end-stage renal disease; HHD, home hemodialysis. * Subgroup of subjects with known initial nephropathy (n = 85). ** whatever the technique used (including in-center hemodialysis).

**Table 2 jcm-12-01357-t002:** Dialysis prescription and operating conditions.

Dialysis Prescription	
Hemodialysis sessions per week (n = 91)	
5	31 (34.5)
6	60 (65.5)
Duration of hemodialysis session (min) (n = 91)	
120	59 (64.8)
150	31 (34.1)
180	1 (1.1)
Blood flow rate (mL/min) (n = 91)	
<250	2 (2.2)
[250–300]	80 (87.9)
[300–350]	9 (9.9)
Dialysate flow rate (mL/min) (n = 85)	
150	2 (2.4)
170	1 (1.2)
180	74 (87.1)
200	8 (9.4)
Dialysate liters per session (n = 78)	
20	4 (5.1)
25	58 (74.4)
30	16 (20.5)
Anticoagulation (n = 91)	
No anticoagulation	19 (20.9)
UFH	6 (6.6)
LMWH	66 (72.5)
Vascular Access (n = 93)	
AV fistula	92 (97.9)
Catheter	2 (2.1)
Vascular access needling	
Buttonhole	83 (89.2)
Rope-ladder rotation	10 (10.8)

Values expressed as n (%), median (IQR). Abbreviations: HHD, home hemodialysis; LMWH, low-molecular-weight heparin; UFH, unfractionated heparin.

**Table 3 jcm-12-01357-t003:** Operating conditions and clinical performances during the follow-up.

Dialysis Performance	Ses. per Week	Total Dialysate Volume	Conv. Volume (SeCoHD)	Weight Loss	Total Dialysate Effluent	Total Dialysate Weekly	Urea K Estimate	Weekly K Urea	Equiv. Urea K	Predialysis Body Weight	TBW Estim	Sdwk KT/V
	N	L/Ses.	L/Ses.	Kg/Ses.	L/Ses	L/Wk	mL/min	L/Wk	mL/min	Kg	L	
N Patients	94	94	94	94	94	94	94	94	94	94	94	94
Median	6.00	21.60	3.00	1.26	26.38	156.07	161	94.00	9.00	71.70	38.60	2.40
Mean	5.65	23.67	3.00	1.27	27.94	156.98	161	94.21	9.17	74.08	39.57	2.52
Std. Deviation	0.480	2.88	1.50	0.46	2.96	13.94	7	8.36	0.81	16.39	9.58	0.67
IQR	1.00	5.40	3.00	0.46	5.40	5.63	7	3.00	0.03	19.35	11.60	0.80
Range	1.00	14.40	4.00	2.39	14.26	91.30	42	55.00	5.00	94.20	53.90	3.40
Minimum	5.00	18.00	2.00	0.25	22.20	127.47	139	76.00	8.00	42.90	21.50	1.10
Maximum	6.00	32.40	6.00	2.64	36.46	218.77	181	131.00	13.00	137.10	75.40	4.50

(Ses., Session; Conv., convective; K, Clearance; TBW, Total Body Water; sdwk, standardized weekly).

**Table 4 jcm-12-01357-t004:** Clinical and laboratory data.

Parameter	Baseline (D0)	]D0–3 mo]	]3–6 mo]	]6–9 mo]	]9–12 mo]	]12–15 mo]	]15–18 mo]	]18–21 mo]	]21–24 mo]
Clinical variables									
Weight loss (g)	1.29 ± 0.46	1.27 ± 0.51	1.28 ± 0.53	1.34 ± 0.57	1.31 ± 0.59	1.33 ± 0.61	1.36 ± 0.55	1.37 ± 0.54	1.40 ± 0.50
N (%)	25 (26.6)	79 (84.0)	80 (85.1)	72 (76.6)	60 (63.8)	51 (54.3)	42 (44.7)	39 (41.5)	32 (34.0)
Predialysis SBP (mmHg)	137.8 ± 20.8	136.4 ± 19.2	132.1 ± 21.1	132.8 ± 20.2	132.6 ± 20.6	135.6 ± 18.3	135.4 ± 20.7	134.9 ± 21.0	138.7 ± 21.2
N (%)	89 (94.7)	70 (74.5)	61 (64.9)	58 (61.7)	48 (51.1)	42 (44.7)	32 (34.0)	32 (34.0)	29 (30.9)
Predialysis DBP (mmHg)	78.4 ± 14.4	75.9 ± 11.8	73.2 ± 11.9	73.9 ± 12.4	73.7 ± 13.2	75.5 ± 11.8	73.9 ± 11.3	74.1 ± 12.2	75.4 ± 12.5
N (%)	89 (94.7)	70 (74.5)	61 (64.9)	58 (61.7)	48 (51.1)	42 (44.7)	32 (34.0)	32 (34.0)	29 (30.9)
Intradialytic hypotensions (%)	14.3	2.7	3.7	3.5	6.5	5.9	3.4	4.9	2.1
N (sessions analyzed)	14	1950	2629	2466	2308	1838	1715	1651	1264
Antihypertensive drugs (%)	39	-	-	-	-	22	-	-	-
N (%)	94 (100.0)	-	-	-	-	50 (53.2)	-	-	-
Biological variables									
Bicarbonate (mmol/L)	22.2 ± 3.5	21.8 ± 3.2	22.1 ± 3.0	21.8 ± 3.1	21.5 ± 2.6	22.3 ± 3.0	21.9 ± 3.1	21.2 ± 3.0	21.6 ± 2.8
N (%)	62 (66.0)	43 (45.7)	48 (51.1)	46 (48.9)	32 (34.0)	34 (36.2)	27 (28.7)	21 (22.3)	17 (18.1)
Phosphate (mmol/L)	1.54 (1.18–1.88)	1.65 (1.33–1.84)	1.66 (1.42–1.85)	1.67 (1.25–2.02)	1.55 (1.39–1.98)	1.54 (1.34–2.02)	1.62 (1.19–1.75)	1.67 (1.46–2.0)	1.55 (1.20–1.89)
N (%)	78 (83.0)	49 (52.1)	53 (56.4)	54 (57.4)	35 (37.2)	37 (39.4)	33 (35.1)	24 (25.5)	19 (20.2)
Calcium (mmol/L)	2.23 ± 0.19	2.20 ± 0.20	2.23 ± 0.17	2.28 ± 0.19	2.30 ± 0.24	2.26 ± 0.24	2.18 ± 0.19	2.24 ± 0.19	2.17 ± 0.17
N (%)	80 (85.1)	51 (54.3)	53 (56.4)	54 (57.4)	35 (37.2)	36 (38.3)	33 (35.1)	24 (25.5)	19 (20.2)
PTH (pg/mL)	306 (139–486)	386 (204–560)	340 (108–575)	368 (197–655)	268 (99–573)	417 (256–745)	434 (329–599)	526 (250–908)	408 (282–738)
N (%)	39 (41.5)	18 (19.1)	19 (20.2)	23 (24.5)	20 (21.3)	20 (21.3)	17 (18.1)	14 (14.9)	10 (10.6)
Hemoglobin (g/dL)	11.38 ± 1.61	10.78 ± 1.55	10.53 ± 1.52	10.85 ± 1.48	10.87 ± 1.50	11.21 ± 1.62	10.69 ± 1.45	10.87 ± 1.56	11.33 ± 1.54
N (%)	74 (78.7)	46 (48.9)	56 (59.6)	50 (53.2)	38 (40.4)	37 (39.4)	32 (34.0)	26 (27.7)	19 (20.2)
Ferritin (µg/L)	160 (102–358)	111 (55–256)	112 (58–210)	125 (49–297)	103 (70–174)	177 (79–350)	219 (65–376)	202 (87–381)	172 (74–407)
N (%)	61 (64.9)	37 (39.4)	46 (48.9)	45 (47.9)	33 (35.1)	30 (31.9)	28 (29.8)	23 (24.5)	16 (17.0)
Transferrin Saturation (%)	24 (17–33)	20 (13–22)	18 (15–24)	19 (13–24)	17 (14–24)	22 (16–29)	23 (17–28)	19 (15–27)	17 (13–31)
N (%)	60 (63.8)	34 (36.2)	44 (46.8)	43 (45.7)	33 (35.1)	30 (31.9)	28 (29.8)	22 (23.4)	16 (17.0)
Albumin (g/L)	40.02 ± 4.44	40.06 ± 5.93	41.60 ± 4.43	41.40 ± 4.53	40.94 ± 4.17	41.87 ± 4.80	40.30 ± 4.39	42.27 ± 4.05	42.17 ± 4.27
N (%)	63 (67.0)	38 (40.4)	48 (51.1)	48 (51.1)	34 (36.2)	33 (35.1)	32 (34.0)	24 (25.5)	18 (19.1)
Creatinine (µmol/L)	773 ± 278	789 ± 284	758 ± 244	772 ± 268	729 ± 293	798 ± 304	766 ± 231	840 ± 292	769 ± 238
N (%)	77 (81.9)	50 (53.2)	53 (56.4)	54 (57.4)	35 (37.2)	33 (35.1)	31 (33.0)	23 (24.5)	19 (20.2)
β2-Microglobulin (mg/L)	23.7 (17.2–30.4)	21.7 (16.4–24.5)	23.7 (21.1–24.9)	20.4 (18.5–29.6)	24.8 (19.3–27.7)	21.4 (18.5–29.4)	24.2 (20.9–33.9)	23.6 (19.8–28.3)	25.4 (20.6–27.8)
N (%)	27 (28.7)	13 (13.8)	14 (14.9)	15 (16.0)	16 (17.0)	15 (16.0)	13 (13.8)	8 (8.5)	6 (6.4)

Values are expressed as median (25–75%) or mean ± SD (mo, months).

**Table 5 jcm-12-01357-t005:** Cardiological data—evolution of the LVM and LVEF during the first 3-month period of follow-up.

	Baseline (D0)	] D0–3 Months]
N	n = 20	n = 20
LVM (g/m²)	114 ± 24	103 ± 27
LVEF (%)	61.5 ± 5.0	62.0 ± 8.5

Values are expressed as n (%), mean ± SD. Abbreviations: LVM, left ventricular mass, LVEF, left ventricular ejection fraction.

## Data Availability

The data presented in this study are available on request from the corresponding author. The data are not publicly available due to ethical concern.

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
