# Peer review of "Two Years’ Experience of Intensive Home Hemodialysis with the Physidia S3 System: Results from the RECAP Study"

_jcm, 2023, doi:10.3390/jcm12041357_

Round 1
Reviewer 1 Report
This real life study is interesting for promoting new machine for doing home dialysis and allows to put forward a technique that seems to be at least equivalent to the more known home dialysis method, the NX stage
However the result and the discussion can be improved and more accurate.
Some figure can be replaced (figure 3/4/5/8)
Table 4 implemented with clinical and biological data evolution during the all Time on S3 would be a major contribution with
- Weight
- PAS/PAD
- Antihypertensive drugs (number)
- Bicarbonate
- Used of Phosphate binder
- EPO dosing
- Iron supplementation
Introduction section:
- the authors must specified that the Nxstage system or S3 system is not IHHD but HHD
Result section
For all table and figure, number of patients described must be specified. Most of the data shown do not appear to be calculated on all 94 patients. number and % must be specified
Some % have to be recalculated. For example line 160/161 54.2+31.3+13.3 = 98.8 but in the figure 2 Underweight patient represent 1%.
- In figure 1 maybe a presentation with number (%) could be more accurate.
- Table 1. Some data should not be available (professional status) and we would like to have the number of data analyzed for each item. Prior dialysis duration: does the data apply to "in center hemodialysis" or the association of the item and peritoneal dialysis and another machine than S3 monitor?
- Table 2. Almost all the results (except vascular access data) are available for 91 patients. What about the 3 missing? Vascular access needling represent 93 patients while 92 have vascular access (one patient used both needling?)
- Figure 3, 4, 5 and 6. Number of patient available for each Time on S3 are mandatory.
- Figure 4. The abscissa legend is missing
- Table 5. Data cannot be exposed in this manner. Comparison of the evolution of LVM and LVF should be done with the same patients between the period (49 patients at D0 and only 20 at D0-3). Line 249/251 are not clear the comparison period for LVM and LVF does not seem the same
- Figure 7. In survival figure, table of number of patient at risk must be added.
- Figure 8. The legend in ordinates is in french
Line 188 to 192 is quite similar to line 177/179
Line 249/251 are not clear the comparison period for LVM and LVF does not seem the same
Line 271 : it is not clear what 42.9% of patients are.
Discussion section
Many of the data described in the discussion are not available in the results presented
This is related to the data that are in table and figure and that are maybe not the most judicious data for this paper.
Line 293/294 : returning to in center hemodialysis observed in 18% ?
Line 318 the very low incidence of intradialytic hypotension are not mentioned in results
Line 321 used of hypertensive medication are not mentioned in result section
Line 329/330 prescription of phosphate binder are not mentioned in result section
Line 334: used of EPO and iron supplementation would have been interesting data
Line 336: data on sodium/potassium and bicarbonate would have been of interest
Line 341/340 data about patient perception and satisfaction would be of interest in the result section
Data of cardiac assessment (line 350/352) : reservations described in the results section
Author Response
Please see the attachment
Manuscript Title: Two Years’ Experience of Intensive Home Hemodialysis with the Physidia S3 System: Results from the RECAP study
Manuscript ID: jcm-2110981
Reply to observations of outside Reviewer 1
We would like to take this opportunity to thank the referee for all her/his comments and help in improving the quality of this article.
A revised manuscript is submitted in which concerns raised could be clarified.
Comments and Suggestions for Authors
This real life study is interesting for promoting new machine for doing home dialysis and allows to put forward a technique that seems to be at least equivalent to the more known home dialysis method, the NX stage
However the result and the discussion can be improved and more accurate.
Some figure can be replaced (figure 3/4/5/8)
Figures 3/4/5/8 have been replaced and improved.
Table 4 implemented with clinical and biological data evolution during the all Time on S3 would be a major contribution with
- Weight : only weight loss was available and has been added in Table 4.
- PAS/PAD : data have been added in Table 4 in addition to Figures.
- Antihypertensive drugs (number) : data, only recorded at baseline and after one year have been added in Table 4.
- Bicarbonates: data have been added in Table 4.
- Use of Phosphate binder: data are unfortunately not available.
- EPO dosing: data are unfortunately not available.
- Iron supplementation: data are unfortunately not available.
Introduction section:
The authors must specify that the Nxstage system or S3 system is not IHHD but HHD
Nxstage system or S3 system correspond to more frequent and /or longer HD modalities which have been defined as intensive HD as reported by Marshall as follows :
“In the literature, more frequent and/ or longer HD is referred to by a plethora of terms, which often focus on the specific prescription enhancement (e.g., “extended-hour,” “nocturnal,” “quotidian,” “daily.” Only 1 term, however, encompasses all the others, that of “intensive” HD (Marshall MR. Intensive hemodialysis-keeping the faith. Kidney Int. 2018 Jan;93(1):10-12. doi: 10.1016/j.kint.2017.09.014. PMID: 29291814.)
This reference has been added in the manuscript as ref 1.
Result section:
For all table and figure, number of patients described must be specified. Most of the data shown do not appear to be calculated on all 94 patients. number and % must be specified
Number and % have been added in the manuscript in Tables and Figures.
Some % have to be recalculated. For example line 160/161 54.2+31.3+13.3 = 98.8 but in the figure 2 Underweight patient represent 1%.
The repartition of the population according to BMI classification was as follows:
1.2% underweight; 54.2% normal; 31.3% overweight; 13.3 % obese corresponding to a total of 100%. In Figure 2, the 1% underweight corresponded to a rounded number down to the nearest whole number.
These data have been included in Table 1 and Figure 2 has been deleted.
- In figure 1 maybe a presentation with number (%) could be more accurate.
The percentages have been added in the Figure.
- Table 1. Some data should not be available (professional status) and we would like to have the number of data analyzed for each item.
Professional status was one of the variables analyzed in the study, the reason why it was reported in Table 1 and initial Figure 8 (new Figure 7). The number of patients described is presented in Table 1 and in Figure 7 (in the legend).
Prior dialysis duration: does the data apply to "in center hemodialysis" or the association of the item and peritoneal dialysis and another machine than S3 monitor?
Prior dialysis duration corresponds to any renal replacement modality whatever the technique used including in-center hemodialysis.
This information has been added in the Legend of Table 1:
Table 1. Patient characteristics.
Values expressed as n (%), mean ± SD, median (25 % – 75 %).
Abbreviation: ESRD, End stage renal disease; HHD, Home HemoDialysis
*Subgroup of subjects with known initial nephropathy (n=85)
** whatever the technique used (including in-center hemodialysis)
- Table 2. Almost all the results (except vascular access data) are available for 91 patients. What about the 3 missing ?
Unfortunately, some of the data were not available for 3 patients of the study.
Vascular access needling represent 93 patients while 92 have vascular access (one patient used both needling?)
Actually, one patient used two vascular access types during the entire study. This has been added in the text line 207:
“Vascular access used was arteriovenous fistula in 97.9 % of patients. One patient used two vascular access types during the entire study”
- Figure 3, 4, 5 and 6. Number of patient available for each Time on S3 are mandatory.
Number of patients has been added in the Figures (new Figures 2, 3, 4 and 5).
- Figure 4. The abscissa legend is missing
Abscissa legend has been added in initial Figure 4 (new Figure 3).
- Table 5. Data cannot be exposed in this manner. Comparison of the evolution of LVM and LVF should be done with the same patients between the period (49 patients at D0 and only 20 at D0-3).
Data have been updated with the only subset of 20 patients.
|
|
Baseline (D0) |
]D0-3 months] |
|
N |
n= 20 |
n=20 |
|
LVM (g/m²) |
114 ± 24 |
103 ± 27 |
|
LVEF (%) |
61.5 ± 5.0 |
62.0 ± 8.5 |
Line 249/251 are not clear the comparison period for LVM and LVF does not seem the same
The remark is correct. This has been revised as follows :
“In a subset of 20 patients, cardiologic features were more precisely monitored by echocardiography during three months of follow-up. As shown in Table 5, left ventricular mass reduced by 9.6% at 3 months (103 vs. 114 g/m²) while ejection fraction remained in the normal range and didn’t change over time (62.0 % vs. 61.5%). “
- Figure 7. In survival figure, table of number of patient at risk must be added.
Number of patients at risk has been added in Figure 7 (new Figure 6).
- Figure 8. The legend in ordinates is in French
Legend of Figure 8 (new Figure 7) has been corrected.
Line 188 to 192 is quite similar to line 177/179
Results have been matched.
Line 249/251 are not clear the comparison period for LVM and LVF does not seem the same The comparison concerns the period M0-M3. The corrections have been done in the text:
“In a subset of 20 patients, cardiologic features were more precisely monitored by echocardiography during three months of follow-up. As shown in Table 5, left ventricular mass reduced by 9.6% at 3 months (103 vs. 114 g/m²) while ejection fraction remained in the normal range and didn’t change over time (62.0 % vs. 61.5%). “
Line 271 : it is not clear what 42.9% of patients are.
The following sentence was confusing and therefore deleted:
“Among the 11 unemployed patients at baseline, a professional activity had been resumed during the follow-up period by 3 patients (42.9% of patients whose resumption of work was documented). 16 patients (8 men and 8. 16 patients (8 men and 8 women) had a period of unemployment during follow-up, and 25% of these patients returned to work at least once during the study period”
Discussion section
Many of the data described in the discussion are not available in the results presented
This is related to the data that are in table and figure and that are maybe not the most judicious data for this paper.
Line 293/294 : returning to in center hemodialysis observed in 18% ?
This has been deleted.
Line 318 the very low incidence of intradialytic hypotension are not mentioned in results
Intradialytic hypotension has been added in Table 4.
Line 321 used of hypertensive medication are not mentioned in result section
Line 329/330 prescription of phosphate binder are not mentioned in result section
Line 334: used of EPO and iron supplementation would have been interesting data
All these mentions have been deleted in the discussion section.
Line 336: data on sodium/potassium and bicarbonate would have been of interest
Only data on bicarbonate have been collected and added in Table 4.
Line 341/340 data about patient perception and satisfaction would be of interest in the result section
These results were previously published by our group and are only discussed here.
This information has been precised in the text line 456:
“ Patient perception and satisfaction survey previously reported by our group confirmed that S3 system was easy to use, easily implemented at home and facilitated patient’s mobility “
Data of cardiac assessment (line 350/352) : reservations described in the results section
Data have been revised in the results section

Reviewer 2 Report
Congratulations on a very nicely written paper. The authors have shared their experience with Physidia, a hemodialysis machine explicitly designed for HHD with less technical complexity than conventional HD machines. The data presented are vital as it reports that HHD with Physidia has a good outcome. I suggest some minor revisions/addition for clarity.
My comments:
- The outcomes measured from the study from the abstract need to be clarified. Addition of one sentence on the outcomes studied in this paper.
- Similarly, in the last paragraph of the introduction, please define the outcome measured in this study.
- Under calculation and analysis, the definition of intradialytic hypotension was sBP < 90 mmHg. Why was this chosen instead of the regular intradialytic hypotension definition?
- Under results and patient characteristics. Please add the typical treatment regimen for in-center dialysis. This is important as we are comparing the outcome at baseline (which is in-center) with Physidia.
- Could figure 2 data be included in Table 1 instead? There are many tables and figures.
- Under discussion, the outcomes (clinical/technique) failure should be compared with other HHD programs that use conventional HD machines. Physician S3 is designed to be more user-friendly than the regular HD machine, but the advantage of the machine will be more evident if the clinical outcomes are comparable to that of conventional HD machines.
Author Response
Please see the attachment
Manuscript Title: Two Years’ Experience of Intensive Home Hemodialysis with the Physidia S3 System: Results from the RECAP study
Manuscript ID: jcm-2110981
Reply to observations of outside Reviewer 2
Comments and Suggestions for Authors
Congratulations on a very nicely written paper. The authors have shared their experience with Physidia, a hemodialysis machine explicitly designed for HHD with less technical complexity than conventional HD machines. The data presented are vital as it reports that HHD with Physidia has a good outcome. I suggest some minor revisions/addition for clarity.
My comments:
- The outcomes measured from the study from the abstract need to be clarified. Addition of one sentence on the outcomes studied in this paper.
Outcomes assessed have been added in the first sentence of the Abstract as follows:
“The RECAP study reports results and outcomes (clinical performances, patient acceptance, cardiac outcomes, technical survival) achieved with the S3 system used as an intensive home hemodialysis (HHD) platform over a three-year French multicenter study”
- Similarly, in the last paragraph of the introduction, please define the outcome measured in this study.
Outcomes assessed through the study are more detailed in the introduction section as follows:
“In this study, we aimed to report results and outcomes (clinical performances, patient acceptance, cardiac outcomes, technical survival) achieved with the S3 system used as an intensive home HD platform over a three-year French multicentric study.”
- Under calculation and analysis, the definition of intradialytic hypotension was sBP < 90 mmHg. Why was this chosen instead of the regular intradialytic hypotension definition?
Whereas several definitions for IDH are available, a nadir systolic blood pressure (<90 mmHg) carries the strongest relation with outcome compared to other definitions, the reason why this definition was used in the methods.
(Sars B, van der Sande FM, Kooman JP. Intradialytic Hypotension: Mechanisms and Outcome. Blood Purif. 2020;49(1-2):158-167. doi: 10.1159/000503776. Epub 2019 Dec 18. PMID: 31851975; PMCID: PMC7114908.)
This reference has been added in the method section:
“Intradialytic hypotension episode was defined as any SBP drop < 90 mmHg measured during the HD session [16] “
- Under results and patient characteristics. Please add the typical treatment regimen for in-center dialysis. This is important as we are comparing the outcome at baseline (which is in-center) with Physidia.
The treatment regimen for in-center dialysis was 3 sessions of 4 hours per week. It has been added in the Table 1 as follows:
|
Prior renal replacement modality (n=94) |
|
|
In-center hemodialysis (3 x 4 hours/week) |
75 (79.8) |
|
Peritoneal dialysis |
2 (2.1) |
|
Kidney transplant |
3 (3.2) |
|
Incident ESKD |
10 (10.6) |
|
HHD with another machine than S3 monitor |
4 (4.3) |
- Could figure 2 data be included in Table 1 instead? There are many tables and figures.
Figure 2 has been deleted and results included in Table 1.
- Under discussion, the outcomes (clinical/technique) failure should be compared with other HHD programs that use conventional HD machines. Physician S3 is designed to be more user-friendly than the regular HD machine, but the advantage of the machine will be more evident if the clinical outcomes are comparable to that of conventional HD machines.
This is a good point raised by the reviewer. Unfortunately, such comparison is not possible in the reported study, since patients treated with Physidia S3 didn’t experience home HD thrice weekly treatment schedule with conventional in-center HD material. In this perspective, one can only compare our long-term technical survival rate (52%) to previous studies [Jayanti A et al. Technique survival in home haemodialysis: a composite success rate and its risk predictors in a prospective longitudinal cohort from a tertiary renal network programme. Nephrol Dial Transplant. 2013;28(10):2612-20 ; Marshall MR, Walker RC, Polkinghorne KR, Lynn KL. Survival on home dialysis in New Zealand. PLoS One. 2014;9(5):e96847] reporting higher technical acceptance (70 to 88%) based on thrice weekly treatment schedule and using conventional in-center HD material. Such discrepancy likely reflects burden of daily treatment and not home treatment with Physidia S3 device.
This comment has been added in the text line 481 as a limitation of the study:
“Fourth, the outcome (clinical/technique) failure should have been compared with other HHD programs that use conventional HD machines. Unfortunately, such comparison was not possible in the reported study, since patients treated with Physidia S3 didn’t experience home HD thrice weekly treatment schedule with conventional in-center HD material. In this perspective, one can only compare our long-term technical survival rate (52%) to previous studies [27,28] reporting higher technical acceptance (70 to 88%) based on thrice weekly treatment schedule and using conventional in-center HD material. Such discrepancy likely reflects burden of daily treatment and not home treatment with Physidia S3 device.“

Round 2
Reviewer 1 Report
the manuscript is clearer than before but there are still some small errors in tables
Table 1 : prior dialysis duration is described for 94 patients while10 are incident ESKD
Table 2 dialysis prescription not described for 94 patients (nearly all the items are done with 91 patients)
I don't really understand why the 3 month cardiological data change beatwwen the two manuscripts
In table 1 professionnal situation is described in 93 patients at D0 whereas in figure 7, 94 patients were reported at inclusion (with exactely the same percentage)
